# Delayed Gastric Emptying Does Not Influence Cancer-Specific Survival after Pancreatoduodenectomy for Pancreatic Ductal Adenocarcinoma

**DOI:** 10.3390/jcm11144200

**Published:** 2022-07-20

**Authors:** Christiane Pillny, Jessica Teschke, Jana Enderes, Steffen Manekeller, Jörg C. Kalff, Tim R. Glowka

**Affiliations:** Department of Surgery, University Hospital Bonn, Venusberg-Campus 1, 53127 Bonn, Germany; christiane.pillny@ukbonn.de (C.P.); jessica.teschke@ukbonn.de (J.T.); jana.enderes@ukbonn.de (J.E.); steffen.manekeller@ukbonn.de (S.M.); joerg.kalff@ukbonn.de (J.C.K.)

**Keywords:** delayed gastric emptying, DGE, pancreatoduodenectomy, cancer-specific survival

## Abstract

Background: Delayed gastric emptying (DGE) remains the most frequent complication following pancreatoduodenectomy (PD). The present study investigates the influence of delayed gastric emptying on cancer-specific survival after PD. Methods: We included 267 patients who underwent PD between 2014 and 2021. They were analyzed regarding demographic factors, pre- and perioperative characteristics, surgical complications, and long-term survival. Results: Patients with a higher Charlson Comorbidity Index (CCI) or pre-existing pulmonary disease suffered significantly more from DGE. When experiencing PPH, a prolonged hospital stay, or major overall complications (Clavien-Dindo °III-V) were more common in the DGE group. Tumor size over 3 cm negatively affected survival. Conclusions: DGE has no influence on long-term survival in PDAC patients, although it prolongs hospital stay.

## 1. Introduction

In Western societies, pancreatic cancer is the fourth leading cause of cancer-related death and is expected to be the second leading cause in 2030 [1]. Surgical resection is the only curative treatment option for PDAC [2]. Although perioperative mortality is low, with 6.1% in tertiary centers in Germany [3], morbidity is still high, with reported rates of up to 30% to 50% [4]. Several studies demonstrate a negative impact of postoperative complications on cancer-specific survival [5,6]. Sandini et al. [7] showed that major complications, seen in up to 19.1% of the patients, independently decreased long-term mortality after PD for PDAC. The most frequent complication after PD is delayed gastric emptying (DGE) with reported incidences of up to 60% [8]. Futagawa et al. showed that clinically relevant DGE following PD negatively affects long-term survival (*p* = 0.03) [9]. Included in the presented Japanese cohort were all entities of periampullary pathologies, although current studies demonstrate that various periampullary entities come along with different long-term survival rates [10].

With a rate of up to 90%, the most frequent malignancy of the pancreas is PDAC [11]. Despite constantly growing knowledge, prognosis still remains poor [12]. Even for patients who underwent resection with curative intent, long-term survival rates could not significantly be improved over recent decades. For initially resectable patients, reported 1- and 2-year survival rates are about 77.9% and 47.4% [13]. Although a recent study reports long term survival of more than 40% in ideal settings [14], most studies report 5-year survival of around 20% [10,15,16]. To exclusively address the effect of DGE following PD on PDAC-related outcome, only patients suffering from PDAC were included in this study.

## 2. Patients and Methods

Between January 2014 and December 2021, a total of 267 patients underwent PD at our department. Of these, 136 underwent surgery for PDAC. All patients who underwent PD due to PDAC were analyzed regarding demographic factors, pre- and perioperative characteristics, surgical complications, and long-term survival. Patients suffering from entities other than PDAC (*n* = 131), or from conditions in which DGE can not be classified (*n* = 3, two patients with longer-lasting mechanical ventilation and one patient with former gastrectomy), were excluded from analysis, so a total of 133 patients were included in the analysis. All pancreatic resections were recorded prospectively in our database after being given written informed consent with approval by the responsible ethical committee (Ethikkommission der Medizinischen Fakultät der Rheinischen Friedrich-Wilhelms-Universität, 347/13). Morbidity and mortality were documented according to the Dindo-Clavien classification [17], with major complications defined as greater or equal to grade 3. Pancreatic fistula (PF), postpancreatectomy hemorrhage (PPH), and DGE were defined according to the classification of ISGPS [8,18,19]. DGE is classified according to the following parameters: requirement of nasogastric tube (NGT), inability to tolerate solid food, vomiting/gastric distension, and use of prokinetics. DGE is then classified into three grades according to the parameters (DGE °A: NGT 4–7 d or reinsertion >POD 3, unability to tolerate solid food by POD 7, vomiting/gastric distension and use of prokinetics facultatively. DGE °B NGT 8–14 d or reinsertion >POD 7, unability to tolerate solid food by POD 14 and vomiting/use of prokinetics obligate. DGE °C NGT >15 d or reinsertion >POD 14, unability to tolerate solid food by POD 21 and vomiting/use of prokinetics obligate).

Adjuvant systemic therapy options were discussed for each patient in our weekly interdisciplinary tumor conference and performed between 4 and 12 weeks after resection, depending on performance and recovery status with FOLFIRINOX (5-FU, Irinotecan and Oxaliplatin) or gemcitabine with or without capecitabine, as recommended in the German and European guidelines [20]. Follow-up was performed in accordance with the mandatory follow-up procedure in our comprehensive cancer center, through clinical examination, serological testing for tumor markers (CEA, CA 19-9), and imaging with CT or MRI. Postoperative survival was recorded by informing the cancer registry of North Rhine Westphalia. Deaths occurring within the postoperative hospital stay (or within 30 days after surgery), were considered as surgical mortality. As previously described, perioperative management was chosen according to an institutional recovery program [21]. Pre-operative sip feeds were not administered on a routine basis, but only in cases of malnutrition. Only when the oral route was inaccessible, was parenteral nutrition administered. Bowel preparation was not performed and pre-operative oral fasting was limited to 2 h for liquids and 6 h for solids. For perioperative pain management, a mid-thoracic epidural catheter was inserted, whereas in the case of contraindications a patient-controlled analgesia (PCA) pump was established. 

PD was carried out with pancreatogastrostomy (PG) by default and a single-loop retrocolic reconstruction, in case of pyloric preservation, as previously described [22]. If a classical whipple procedure with antral resection was performed, due to infiltration into the distal stomach, reconstruction was either performed as a single-loop-duodenoenterostomy (DE) or as a BII- or ReY-reconstruction, depending upon the preference of the surgeon. All patients spent at least one night in our intensive care or intermediate care unit. A 14 French NGT was placed intraoperatively and was removed when its output was below 500 mL/day. Two soft drains that were placed beside the PG and BDA were removed between postoperative days 3 and 5 if amylase content was not elevated (compared to serum amylase). DGE was recorded according to the ISGPS definition and, therefore, classified into three grades (A–C) based on duration of NGT, need for reinsertion, the day when solid food was first tolerated, occurrence of vomiting, and use of prokinetics. Data were analyzed regarding histological characteristics, such as: tumor size; differentiation; resection margins; lymph-node, microlymphatic, microvascular, and perineural invasion; resection margin; and UICC stage.

### Statistical Analyses

Data were recorded with Excel 2016 (Microsoft Corporation, Redmond, Washington, DC, USA) and analyzed with SPSS 27 (IBM Corporation, Armonk, New York, NY, USA). Continuously and normally distributed variables were expressed as medians ± standard deviation and analyzed using Student’s *t* test, while non-normally distributed data were expressed as medians and interquartile ranges and analyzed using the Mann–Whitney *U* test. Categorical data were expressed as proportions and compared with the Pearson *χ*^2^ or the Fisher’s exact test, as appropriate. Factors with *p* < 0.1 in the univariate analysis were included in multivariate stepwise logistic regression analysis. The relative risk was described by the estimated odds ratio with 95% confidence intervals. A *p*-value < 0.05 was considered statistically significant.

Survival curves were estimated using the Kaplan–Meier method. Cox regression analysis was performed to identify predictors for long-term survival.

## 3. Results

Of 267 patients undergoing pancreatoduodenectomy in the study period, 133 were treated for PDAC. Of these, 31 patients (23%) suffered from clinically relevant DGE (°B/C), while 102 patients did not (DGE °0/°A). Both groups were comparable regarding demographics, as shown in Table 1, apart from patients suffering from chronic pulmonary disease. Patients with preexisting pulmonary disease were suffering statistically more often from clinically relevant DGE than patients without preexisting pulmonary disease (29% vs. 7.8%, *p* = 0.005). Patients with a higher Charlson Comorbidity Index (CCI) suffered significantly more from DGE in comparison with patients not suffering from clinically relevant DGE (*p* = 0.031). Intraoperative characteristics were largely evenly distributed between the two groups (Table 2). Especially technical aspects, such as the choice of reconstruction, the extent of lymphadenectomy, or the need for vascular reconstruction, did not influence the frequency of DGE. Surgical complications, especially (clinically relevant) PF, did not significantly differ between patients suffering from DGE and those who did not. Nevertheless, regarding PPH, statistically more patients suffering from PPH experienced clinically relevant DGE (31.4% vs. 51.6%, *p* = 0.04), which is consistent with clinically relevant PPH °B/C (24.5% vs. 45.2%, *p* = 0.027). Furthermore, major overall complications were more common in the DGE group (Clavien-Dindo °III-V 38.2% vs. 87.1%, *p* < 0.001) and hospital stay was significantly prolonged (18 vs. 29 days, *p* < 0.001).

To rule out tumor-related effects on oncological survival, histopathological factors were compared, as shown in Table 3. Neither tumor size; lymph node metastases; microlymphatic-, microvascular-, or perineural invasion; nor positive surgical margins were different in patients with or without DGE.

Risk factors for impaired median survival are shown in Table 4 and Figure 1. Intra-operative characteristics did not significantly lead to a difference in survival. Tumor size over 3 cm negatively affected survival (11 vs. 26 months; *p* = 0.003). The frequency of postoperative complications, including clinically relevant DGE, (clinically relevant) PF and (clinically relevant) PPH, had no impact on survival. In a multivariate regression analysis (Table 5) clinically relevant PPH (°B/C) acts as an independent factor shortening survival (*p* = 0.024), as well as medical complications were confirmed to shorten median survival (*p* = 0.009).

## 4. Discussion

Survival rates in patients suffering from PDAC remain poor, even in patients undergoing curative surgical treatment [23]. Several studies on postoperative complications following PD for pancreatic carcinoma suggested a negative impact on patient survival rates [5,6]. However, precise data on the impact of DGE on overall survival are still scarce. 

In a recent study from Japan [9], Futagawa and colleagues demonstrated a significant impact of DGE on overall survival. A potential restriction of the cited study was the inclusion of different periampullary pathologies, although it is known that the primary disease itself has a tremendous impact on patient survival [10]. The other question was whether or not the results from the Japanese cohort would also be valid with a German cohort, as we know that incidence and outcome of tumor entities can differ considerably between the continents [24]; thus we analyzed the effect of clinically relevant DGE on oncological long-term survival in our patients. A total of 133 patients were included in the presented study, all of whom were operated upon for PDAC between January 2014 and December 2021. Of those, 49.6% suffered from major complications (Dindo-Clavien ≥ °III). Regarding DGE, significantly more patients suffering from DGE showed a higher rate of major complications, as measured by Dindo-Clavien °III to °V. The rate of clinically relevant DGE °B/C in our cohort was 23.3% and its occurrence did not show any influence on overall, cancer-specific survival (*p* = 0.916). The survival curve with DGE was almost identical to the survival curve without DGE, therefore DGE does not influence cancer-specific survival for PDAC. Looking at other tumor entities, postoperative complications are proven to worsen postoperative overall survival. For colorectal cancer, studies have shown close correlation of anastomotic leakage on decreasing survival rates [25].

Pancreatic fistula is the most feared major complication following PD. For pancreatic surgery, recent data are controversial with regard to whether PF acts as an independent risk factor or not [26,27,28]. In our cohort, PF, like DGE, could not be identified as a risk factor for shorter survival. This is in line with other reports, which do not demonstrate any effect of relevant complications on long term-survival [29]. Clinically relevant PPH acts as a risk factor for clinically relevant DGE, according to our analysis: 45.2% of the patients with postoperative DGE also suffered from postoperative pancreatic hemorrhage. A correlation of PPH with DGE was already shown [21]. Eventually, pancreatogastrostomy, as performed in our department by default, with bleeding often occurring from the pancreatic remnant directly to the stomach, might be an explanation for DGE, as PPH is more common following PG [4]. However, a correlation of DGE and PPH following pancreatojejunostomy was also described [30,31]. The reason is unknown; both parameters cannot be observed in isolation, in a clinical setting. DGE significantly prolongs hospital stay (29 vs. 18 days, *p* = <0.001), which has been shown several times before [32], but this prolonged recovery does not impact long-term survival. This is in line with a former Dutch study: 33% of all patients having received surgical resection of PDAC did not receive adjuvant chemotherapy [33], presumably due to prolonged recovery. Risk factors for not receiving adjuvant treatment in this study were: higher age, low annual surgical volume and, most importantly, surgical complications such a pancreatic fistula or postpancreatectomy hemorrhage. Clinically relevant DGE, however, did not show any influence on receiving adjuvant treatment. Nussbaum and colleagues showed in a US setting that even fewer patients, 52.7% (open PD) and 55.3% (minimally invasive PD), respectively, received adjuvant chemotherapy [34]. Adjuvant chemotherapy can provide excellent outcomes: an overall survival rate at 3 years of 63.4% and a median disease-free survival of 21.6 months for FOLFIRINOX chemotherapy have been described [14]. Even in a neoadjuvant setting of borderline resectable PDAC, chemotherapy use and the number of chemotherapy cycles were both independent factors for survival [35]. However, as stated above, DGE does not influence the utilization of adjuvant chemotherapy. Our results showing pre-operative chronic pulmonary disease as a risk factor for developing DGE are in line with former findings [36,37]. In contrast to that, own data showed a decrease in DGE after PD in active smoking patients [37]. Studies evaluating the effect of transcutaneous administration of nicotine on DGE development are in preparation in the authors’ department. Actually, treatment for DGE mainly focuses on prophylactic measurements. Table 6 gives an overview about factors known to influence the development of DGE.

Further factors lately proven to affect overall survival after PD for PDAC are histological features, such as tumor size [29], positive surgical margin, and lymphatic invasion. Long-term survival was evenly distributed among patients with and without DGE, demonstrating no effect of DGE on survival in our German cohort. As tumor-related histopathological factors cannot be influenced, other strategies are needed to improve survival for patients suffering from PDAC. PPH, and especially factors preventing PPH, should be considered for further examination, as it has been shown to produce a statistically significant improvement in long-term survival in our patients. Minimally invasive surgery was able to shorten hospital stay, but so far without an effect on overall survival [34,45]. The present study has the known shortcomings of retrospective analyses, although data are prospectively recorded in our pancreatic resection database. Creating prospective randomized evidence with the existing question (cancer-specific survival according to DGE status), is hard to create, as it is not known, before surgery, which patient will develop DGE. At the authors’ department, we are currently prospectively evaluating the effect of minimally invasive surgery on overall survival for PDAC. Our study underlines that DGE significantly worsens patients’ comfort and prolongs hospital stay, but has no impact on long-term survival in PDAC patients.

## Figures and Tables

**Figure 1 jcm-11-04200-f001:**
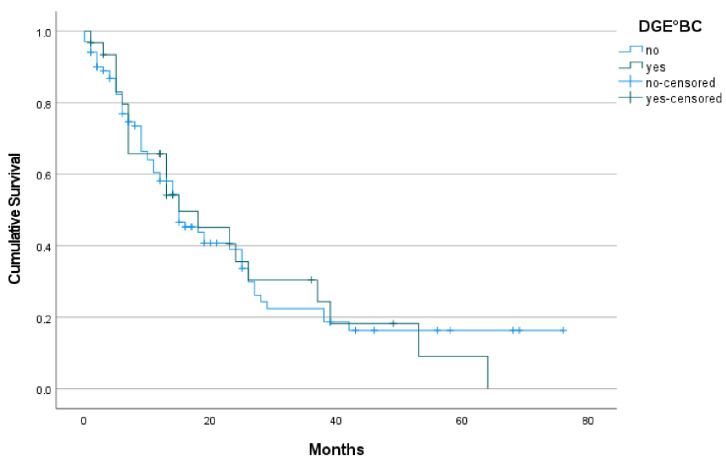
Kaplan–Meier survival function DGE.

**Table 1 jcm-11-04200-t001:** Demography/patient cohort characteristics.

	No DGE	Clinically Relevant DGE (°B/C)	*p* Value
Variables	*n* = 102	*n* = 31	
Female	51 (50%)	12 (38.7%)	0.27
Age > 75 years	36 (35.3%)	11 (35.5%)	0.985
Active Smoker	28 (27.5%)	5 (16.1%)	0.175
Smoker	51 (50%)	12 (38.7%)	0.337
Alcohol consumption	30 (29.4%)	10 (32.3%)	0.729
Pre-operative weight loss	65 (63.7%)	17 (54.8%)	0.407
Pre-operative biliary stenting	53 (52%)	15 (48.4%)	0.727
Pre-existing diabetes mellitus	40 (39.2%)	10 (32.3%)	0.484
Chronic pulmonary disease	8 (7.8%)	9 (29%)	0.005
Previous abdominal surgery	56 (5.9%)	16 (51.6%)	0.755
Charlson Comorbidity Index	3 (2–3)	3 (2–4)	0.031
Exitus < 30 days	5(4.9%)	1 (3.2%)	0.573

Data are expressed as mean ± SD, number (%), or median (interquartile range).

**Table 2 jcm-11-04200-t002:** Perioperative characteristics.

	No DGE	Clinically Relevant DGE (°B/C)	*p* Value
Variables	*n* = 102	*n* = 31	
Hospital stay postoperative	18 (14–24)	29 (25–37)	<0.001
ICU stay	2 (1–3)	1 (1–5)	0.911
Intraoperative blood loss (mL)	700 (400–1100)	600 (350–1000)	0.563
Single loop reconstruction	83 (81.4%)	28 (90.3%)	0.24
Antecolic reconstruction	18 (17.6%)	2 (6.5%)	0.155
Multivisceral resection	7 (6.9%)	2 (6.5%)	1.0
Pylorus preservation	80 (78.4%)	27 (87.1%)	0.287
Extended lymphadenectomy	63 (61.8%)	20 (64.5%)	0.782
Portal venous resection	31 (30.4%)	6 (19.4%)	0.219
Bile duct > 5 cm	30 (29.4%)	4 (12.9%)	0.099
PPH	32 (31.4%)	16 (51.6%)	0.04
Clinically relevant PPH (°B/C)	25 (24.5%)	14 (45.2%)	0.027
PF	17 (16.7%)	5 (16.1%)	0.927
Clinically relevant PF (°B/C)	4 (3.9%)	4 (12.9%)	0.087
Surgical site infection	16 (15.7%)	5 (16.1%)	1.0
Intra-abdominal collection/abscess	9 (8.8%)	3 (9.7%)	1.0
Bacterobilia	54 (52.9%)	14 (45.2%)	0.388
Clavien-Dindo stage Major (°III–V)	39 (38.2%)	27 (87.1%)	<0.001

Data are expressed as mean ± SD, number (%), or median (interquartile range). ICU: intermediate care unit, PPH: postpancreatectomy hemorrhage, PF: pancreatic fistula.

**Table 3 jcm-11-04200-t003:** Histological features.

	No DGE	Clinically Relevant DGE (°B/C)	*p* Value
Variables	*n* = 102	*n* = 31	
Tumor size (cm)	3.15 (2.5–4.2)	2.75 (2.32–3.5)	0.051
Tumor size > 3 cm	59 (57.8%)	12 (38.7%)	0.129
Lymph node metastasis	79 (77.5%)	21 (67.7%)	0.352
Microlymphatic invasion	38 (37.3%)	13 (41.9%)	0.695
Microvascular invasion	19 (18.6%)	6 (19.4%)	0.965
Perineural invasion	72 (70.6%)	23 (74.2%)	0.617
Surgical margin positive	30 (29.4%)	8 (25.8%)	0.748

**Table 4 jcm-11-04200-t004:** Overall survival, Kaplan–Meier analysis.

Variables	No. of Patients	Median Survival (Months)	*p* Value
Pre-operative biliary stenting			0.988
yes	68	15	
no	65	18	
Bacterobilia			0.677
yes	68	15	
no	59	15	
Former or active nicotine consumption			0.969
yes	63	18	
no	65	15	
Active Nicotine consumption			0.826
yes	33	15	
no	97	15	
DM with TOD			0.004
yes	6	7	
no	126	15	
Portal venous resection			0.108
yes	37	10	
no	95	19	
DGE			0.376
yes	67	15	
no	66	19	
Clinically relevant DGE (°B/C)			0.916
yes	31	15	
no	102	15	
Clinically relevant PF (°B/C)			
yes	8	64	0.458
no	124	15	
Clinically relevant PPH (°B/C)			0.881
yes	39	15	
no	94	15	
PPH			0.587
yes	48	15	
no	85	15	
Lymph node metastasis			0.174
yes	100	15	
no	31	18	
Positive surgical margin			0.006
yes	93	14	
no	38	18	
Tumor size >3 cm			
yes	71	11	0.003
no	57	26	

DGE: delayed gastric emptying, PF: pancreatic fistula, PPH: postpancreatectomy hemorrhage, DM: Diabetes mellitus, TOD: target organ damage.

**Table 5 jcm-11-04200-t005:** Multivariate survival analysis, Cox regression.

	Odds Ratio	95% CI	*p*
Clinically relevant PPH (°B/C)	4.802	1.092–21.113	0.024
Medical complications	6.445	1.247–33.307	0.009

CI confidence interval.

**Table 6 jcm-11-04200-t006:** Factors influencing the frequency of DGE.

Study	Number of Patients	Study Type	Parameter Effecting DGE	Effect
**Reconstruction Techniques**				
Hüttner et al., 2022 [38]	650	CDR	Ante-/retrocolic position	ante = retro
Klaiber et al., 2018 [39]	992	Meta	Pylorus preservation	pp = pr
Klaiber et al., 2015 [40]	802	Meta	Single/dual-loop	single = dual
**Risk Factors**				
Park et al., 2009 [41]	129	ROT	Pancreatic fistula	more DGE
Kunstman et al., 2012 [42]	235	ROT	Intraoperative blood loss	more DGE
Hafke et al., 2020 [22]	138	ROT	Supra-/infracolic DE	No effect
Enderes et al., 2021 [37]	295	ROT	Active smoking	less DGE
Enderes et al., 2021 [43]	275	ROT	Liver cirrhosis	no effect
Enderes et al., 2022 [44]	211	ROT	Obesity	no effect

CDR = Cochrane database review, DE = duodenoenterostomy, Meta = meta analysis, pp = pylorus preservation, pr = pylorus resection, RCT = randomized controlled trial, and ROT = retrospective observation trial.

## Data Availability

Our anonymized patient database still contains protected health care information, with which certain patients could be identified (e.g., date of surgery). According to German law these data must not be published. Access to the database can be obtained from the corresponding author upon reasonable request.

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
