# Peer review of "Delayed Gastric Emptying Does Not Influence Cancer-Specific Survival after Pancreatoduodenectomy for Pancreatic Ductal Adenocarcinoma"

_jcm, 2022, doi:10.3390/jcm11144200_

Round 1

Reviewer 1 Report

I would like the authors to present a table with several publications with cases with DGE after pancreatic surgery with the problems of diagnosis, treatment, type of interventions and survival. Also, the conclusions must be more exhaustive and appear in the article not only in the abstract.

The International Study Group of Pancreatic Surgery classifications for DGE.

I would like to talk more about factor involved in the developing of postoperative pancreatic fistulas. such as the association of adenocarciomas with areas of acute pancreatitis has led to a significant increase in postoperative pancreatic fistula and DGE.

Author Response

We thank for the valuable comments. We tried to meet the suggestions given by the reviewers as concise as possible.

Reviewer 1

- I would like the authors to present a table with several publications with cases with DGE after pancreatic surgery with the problems of diagnosis, treatment, type of interventions and survival. Also, the conclusions must be more exhaustive and appear in the article not only in the abstract.

Diagnosis of DGE is clearly defined since the 2007 ISGPS definition. According to your next comment, we inserted a table explaining the ISGPS definition in detail. The ISGPS definition has been critically evaluated and found useful for grading DGE (PMID 20632270).

DGE is thought to reduce patient comfort and prolong hospital stay but is not life-threatening and self-limiting (PMID 17021788). This might be one reason, why no evidence-based treatment for DGE exists. Several papers identified risk factors for the development for DGE and tried to draft prophylactic strategies. An overview is now given in Table 7. In the department of the authors, a standardized treatment strategy is the insertion of a jejunal feeding tube with gastric venting valve (“Trelumina” tube) for prolonged DGE °B/C (manuscript in preparation). A prospective study evaluating transcutaneous nicotine administration is in preparation.

The conclusions of the papers have been stated more prominent in the discussion section, also the main conclusion of the manuscript is the last line of our manuscript.

- The International Study Group of Pancreatic Surgery classifications for DGE.

The ISGPS-definition and –grading of DGE has been inserted in the Patients&Methods section for clarification.

I would like to talk more about factor involved in the developing of postoperative pancreatic fistulas. such as the association of adenocarciomas with areas of acute pancreatitis has led to a significant increase in postoperative pancreatic fistula and DGE.

Pancreatic fistula (POPF) is not the focus of this paper. POPF is clearly a risk factor for the development of (secondary) DGE and pancreatitis is a risk factor for the development of both POPF and DGE as well (see below).

A prospective evaluation of risk factors, outcome and treatment of POPF following pancreatoduodenectomy in our tertiary department is currently under evaluation in another journal. Frequency of POPF °B/C was 19.5%. Risk factors are higher BMI (P=0.002), benign diagnosis (e.g. pancreatitis, P<0.001), soft pancreatic parenchyma (P<0.001). Intraabdominal abscess formation and DGE °B/C are much more common in the presence of POPF (P=0.009 and P<0.001 respectively). Please understand that POPF cannot be discussed more in detail in this manuscript, especially as POPF does not influence cancer-specific survival.

Reviewer 2 Report

Thank you for your manuscript.

Could you please make the inclusion patient or population of the study more clear. Did you include all patients after PD 267 patients?

while in methods you only included PD due to PDAC, and in result only 133 patients.

Abstract: The present study investigates the influence of delayed gastric emptying on cancer-specific survival after PD.

Patients & Methods : All patients who underwent PD due to PDAC , Patients suffering from entities other than PDACwere excluded.

(Line 41-43) “While recent studies reprt long term”, could you please cite more studies in this section.

(Method line 49), Patients suffering from entities other than PDAC were excluded from analysisplease explain the reasons, since you mentioned including total 267 patients underwent PD, while the number of PDAC patients 133 (result line 103). 

(Line55-56): postpancreatectomy hemorrhage (POPF), maybe you mean (PPH), POPF refer to postoperative Pancreatic fistula.

(Line 142) “Table 4. overall survival, Kaplan-Meier analysisplease explain why the No. of patients in most of the variables are more than the including number in the study 133 patients. 

(Line 159) A total of 267 patients could be included in the presented study, all of whom were operated for PDAC between January 2014 and December 2021.

(Line 173-174) “Clinically relevant PPH acts as a risk factor for clinically relevant DGE”, please provide more discussion 

(Line 173-174) “Minimally invasive surgery was able to shorten hospital stay” 

What was the number of patients in minimally invasive surgery and open surgery? And P value? Perioperative characteristics

please provide the related limitation of the study.

 Thanks

Author Response

We thank for the valuable comments. We tried to meet the suggestions given by the reviewers as concise as possible.

Could you please make the inclusion patient or population of the study more clear. Did you include all patients after PD 267 patients?

while in methods you only included PD due to PDAC, and in result only 133 patients.

We very much apologize for the confusion. In the study period, 267 pancreatoduodenectomies (PD) were carried out at our department. Of these patients, 136 underwent surgery for PDAC. Three patients were excluded from analysis (two patients with longer lasting mechanical ventilation, therefore DGE classification was not possible and one patient after former gastrectomy). Patient numbers are clarified as well in the Patients & Methods section as well as in Results.

Abstract: The present study investigates the influence of delayed gastric emptying on cancer-specific survival after PD.

Patients & Methods : All patients who underwent PD due to PDAC , Patients suffering from entities other than PDACwere excluded.

(Line 41-43) “While recent studies reprt long term”, could you please cite more studies in this section.

Only the PRODIGE study reported a long-term survival of more than 40% in the FOLFIRINOX group. This can be explained by the fact, that only “best survivors” entered the study (randomization AFTER surgery), so all patients suffering from postoperative complications or otherwise disqualified for adjuvant treatment were excluded. We changed L41 from plural to singular. Most studies report worse long-term survival, mostly around 20-25%. CONKO-5 and ESPAC-4 trials have been added (L43).

(Method line 49), Patients suffering from entities other than PDAC were excluded from analysis, please explain the reasons, since you mentioned including total 267 patients underwent PD, while the number of PDAC patients 133 (result line 103).

Please see above: “In the study period, 267 pancreatoduodenectomies (PD) were carried out at our department. Of these patients, 136 underwent surgery for PDAC. Three patients were excluded from analysis (two patients with longer lasting mechanical ventilation, therefore DGE classification was not possible and one patient after former gastrectomy). 133 patients entered analysis. Patient numbers are clarified as well in the Patients & Methods section as well as in Results.”

(Line55-56): postpancreatectomy hemorrhage (POPF), maybe you mean (PPH), POPF refer to postoperative Pancreatic fistula.

Of course, you are right, please apologize for the mistake. POPF has been changed to PPH.

(Line 142) “Table 4. overall survival, Kaplan-Meier analysis”. please explain why the No. of patients in most of the variables are more than the including number in the study 133 patients.

Please see above. Of 267 patients, 136 underwent surgery for PDAC. Three more patients were excluded, since DGE could not be measured in these three. 133 entered univariate analysis. Unfortunately, Kaplan-Maier-analysis was carried out with the above mentioned 136 patients, this is why in certain parameters patient numbers add up to 136. This was corrected and complete survival analysis was carried out again with the 133 patients. (Lower numbers than 133 are due to missing values (e.g. in only 128 smoking habits were recorded)). No changes in the significance of the parameters and the general massage occurred with the new analysis. We apologize for the mistake.

(Line 159) A total of 267 patients could be included in the presented study, all of whom were operated for PDAC between January 2014 and December 2021.

The correct number reads 133. This was corrected.

(Line 173-174) “Clinically relevant PPH acts as a risk factor for clinically relevant DGE”, please provide more discussion

PPH was significantly associated with DGE in the current analysis, but also in historical cohorts at our department. The increased rate following PG and evidence for the correlation following PJ anastomosis were provided.

(Line 173-174) “Minimally invasive surgery was able to shorten hospital stay”

What was the number of patients in minimally invasive surgery and open surgery? And P value? Perioperative characteristics

This sentence relates to two studies showing shorter hospital stay without improvement of oncological survival (Refs 34, 45). Nussbaum et al. reported a shorter hospital stay of MIPS vs. open surgery of -2,28 days (P=0.07). Croome showed 6 (MIPS) vs. 9 days (open, P<0.001). We are currently evaluating the effect of robotic PD on oncological survival in our department. Follow-up is not complete, so I cannot provide you with statistics yet.

please provide the related limitation of the study.

The shortcomings of our study (retrospective design) are now mentioned in the discussion section.

Round 2

Reviewer 1 Report

I agree this variant of the Article. Congratulations of the authors!